# Learning from Bad Data via Generation

**Tianyu Guo[1,2,*] Chang Xu[2,*], Boxin Shi[3,4,*], Chao Xu[1], Dacheng Tao[2]**
[1]Key Laboratory of Machine Perception (MOE), CMIC, School of EECS,
Peking University, 100871, China
[2]UBTECH Sydney AI Centre, School of Computer Science, Faculty of Engineering,
The University of Sydney, Darlington, NSW 2008, Australia
[3]National Engineering Laboratory for Video Technology,
Department of Computer Science and Technology, Peking University, Beijing, 100871, China
[4]Peng Cheng Laboratory, Shenzhen, 518040, China
{tianyuguo, shiboxin}@pku.edu.cn, chaoxu@cis.pku.edu.cn
{c.xu, dacheng.tao}@sydney.edu.au

## Abstract

Bad training data would challenge the learning model from understanding the underlying data-generating scheme, which then increases the difficulty in achieving satisfactory performance on unseen test data. We suppose the real data distribution lies in a distribution set supported by the empirical distribution of bad data. A worst-case formulation can be developed over this distribution set, and then be interpreted as a generation task in an adversarial manner. The connections and differences between GANs and our framework have been thoroughly discussed. We further theoretically show the influence of this generation task on learning from bad data and reveal its connection with a data-dependent regularization. Given different distance measures (*e.g.*, Wasserstein distance or JS divergence) of distributions, we can derive different objective functions for the problem. Experimental results on different kinds of bad training data demonstrate the necessity and effectiveness of the proposed method.

## 1 Introduction

Machine learning techniques are applied to fit the data distribution induced by the training set and then make predictions for new examples in various applications, such as image classification [18, 37, 39, 20], image generation [22, 40, 35, 14], and semantic segmentation [12, 36, 31, 17]. An important assumption underlying the success of these methods is that the training set and the test set are subject to the same distribution. It is therefore expected that the models well trained on the training set can also achieve similar performance on the test data that have never been seen before in the training set.

The true underlying distribution of the data is unknown and many methods can be applied to approximate it. For example, cross-entropy loss is often taken as the objective function of deep neural networks in classification tasks, which is equivalent to a maximum likelihood estimation of the unknown data distribution based on the training data [13]. However, many factors, such as the size of training set [13], the way data is collected [11], and the balance between different categories in the training [30], will affect the results of maximum likelihood estimation. If the data distribution approximated by the well trained model on the training sample is far from true data distribution, performance on the test set would then hardly be comparable with that on the training set.

In real-world applications, there usually exist "bad" data that are instantiated from the imbalanced, noisy or reduced training set, resulting in settings where the observed training samples do not well

---

represent the true underlying distribution of the data. In this paper, we study the problem of learning from bad data and propose an adversarial learning strategy based on our theoretical analysis. Instead of optimizing risk under the uniform empirical distribution over the observed bad data as classical methods, we turn to an expected loss against a family of distributions that could contain the true data-generating scheme with high confidence. Specifically, a deep neural network is introduced to approximate the latent data distribution characterized by the observed bad data and the properties of true data distribution. Given Wasserstein distance as the measurement between distributions, we establish a three-player game to solve a worst-case problem. We provide theoretical analysis to show that the optimal generator captures the observed empirical distribution and fits a worse distribution for the classifier. The proposed method roughly corresponds to a data-dependent gradient regularization over the empirical distribution, and we provide a performance guarantee for optimization on the learned distribution. Experiments on multiple natural image datasets confirm that the proposed method provides a robust approach to complement bad training data in different scenarios.

## 2 Proposed Method

Consider a training set $X = \{(x_1, y_1), \cdots, (x_m, y_m)\}$ containing $m$ examples, which are independently sampled from an unknown data distribution. Ideally, the empirical distribution $\hat{\mathbb{P}}_N$ is a good estimation of the true distribution $\mathbb{P}_N$, which means that parameters learned on the empirical distribution will eventually converge to values learned on the true distribution. However, there is a certain distance between the empirical distribution $\hat{\mathbb{P}}_N$ and the real distribution $\mathbb{P}_N$ in practice. This results in unsatisfactory performance of the test samples obtained by the model learned from the empirical distribution. The discrepancy between empirical distribution and real distribution could be caused by many reasons, *e.g.*, samples are polluted by noise or samples of some categories are hard to obtain which reduces the number of samples in particular categories. To restore the performance of models on bad data, we need to reflect the conventional empirical risk minimization over the training data.

Suppose that the real data distribution is in an ambiguity set supported by the empirical distribution $\hat{\mathbb{P}}_N$. We thus propose to optimize an upper bound of the loss function over all probability distributions in this ambiguity set,

$$\inf_{\boldsymbol{\theta} \in \Theta} \sup_{\mathbb{Q} \in \mathbb{B}_\epsilon(\hat{\mathbb{P}}_N)} \mathbb{E}_{\mathbb{Q}}[\ell_\theta(\boldsymbol{x}, y)], \tag{1}$$

where the distribution set $B_\epsilon(\cdot)$ contains all the distributions $\mathbb{Q}$ whose distance from the empirical distribution $\hat{\mathbb{P}}_N$ does not exceed $\epsilon$. The distribution set $B_\epsilon(\cdot)$ is defined as follows

$$\mathbb{B}_\epsilon(\hat{\mathbb{P}}_N) \triangleq \{\mathbb{Q} \in \mathcal{M}(\mathcal{Z}) : d(\mathbb{Q}, \hat{\mathbb{P}}_N) \leq \epsilon\}, \tag{2}$$

where $d(\cdot, \cdot)$ stands for some pre-defined distance metric, $\mathcal{M}(\mathcal{Z})$ denotes the set of probability measures supported on $\mathcal{Z}$, and $\mathcal{Z}$ is the set of possible values of $(\boldsymbol{x}, y)$. According to this definition, we will investigate all possible distributions within a ball centred at $\hat{\mathbb{P}}_N$ with radius $\epsilon$. We aim to discover a distribution $\mathbb{Q}$ from $\mathbb{B}_\epsilon(\hat{\mathbb{P}}_N)$ that corresponds to the worst case, and thus the optimization over this worst case distribution would imply an optimization over the entire distribution set $\mathbb{B}_\epsilon(\hat{\mathbb{P}}_N)$, which also assum to includes real data distribution $\mathbb{P}_N$.

Eq. (1) is intractable, as the worst-case distribution $\mathbb{Q}$ is unknown. As a result, in the following we focus on the inner part of Eq. (1) to find the worst-case distribution $\mathbb{Q}$. Firstly, we re-express the inner part of our objective function defined in Eq. (1) as follows,

$$\sup_{\mathbb{Q} \in \mathbb{B}_\epsilon(\hat{\mathbb{P}}_N)} \mathbb{E}_{\mathbb{Q}}[\ell_\theta(\boldsymbol{x}, y)] = \begin{cases} \sup\limits_{\Pi, \mathbb{Q}} & \int_{\mathcal{Z}} \ell_\theta(\boldsymbol{x}, y) \mathbb{Q}\left(\mathrm{d}(\boldsymbol{x}, y)\right) \\ \text{s.t.} & d(\mathbb{Q}, \hat{\mathbb{P}}_N) \leq \epsilon. \end{cases}, \tag{3}$$

$\Pi$ refers to a joint distribution of $(\boldsymbol{x}, y)$ and $(\boldsymbol{x}', y')$ with marginals $\mathbb{Q}$ and $\hat{\mathbb{P}}_N$ respectively. With the help of standard duality augment, we have

$$\sup_{\mathbb{Q} \in \mathbb{B}_\epsilon(\hat{\mathbb{P}}_N)} \mathbb{E}_{\mathbb{Q}}[\ell_\theta(\boldsymbol{x}, y)] = \sup_{\mathbb{Q} \in \mathcal{M}(\mathcal{Z})} \inf_{\lambda \geq 0} \int_{\mathcal{Z}} \ell_\theta(\boldsymbol{x}) \mathbb{Q}\left(\mathrm{d}(\boldsymbol{x}, y)\right) + \lambda \cdot \left(\epsilon - d(\mathbb{Q}, \hat{\mathbb{P}}_N)\right)$$

$$\leq \inf_{\lambda \geq 0} \sup_{\mathbb{Q}_i \in \mathcal{M}(\mathcal{Z})} \left\{\lambda \epsilon + \int_{\mathcal{Z}} \ell_\theta(\boldsymbol{x}, y) \mathbb{Q}\left(\mathrm{d}(\boldsymbol{x}, y)\right) - \lambda \cdot d(\mathbb{Q}, \hat{\mathbb{P}}_N)\right\} \tag{4}$$

$$= \inf_{\lambda \geq 0} \left\{\lambda \epsilon + \sup_{\mathbb{Q}} \int_{\mathcal{Z}} \ell_\theta(\boldsymbol{x}, y) \mathbb{Q}\left(\mathrm{d}(\boldsymbol{x}, y)\right) - \lambda \cdot d(\mathbb{Q}, \hat{\mathbb{P}}_N)\right\},$$

where $\lambda$ is Lagrangian multiplier. The first term in Eq. (4) is independent from the distribution $\mathbb{Q}$ and the loss function, so we get the constraint that the distribution $\mathbb{Q}$ should satisfy as following,

$$\sup_{\mathbb{Q}}[\mathbb{E}_{\mathbb{Q}}[\ell_\theta(\boldsymbol{x}, y)] - \lambda \cdot d(\mathbb{Q}, \mathbb{P})]. \tag{5}$$

It can be seen that the distribution $\mathbb{Q}$ should make the loss function as large as possible while trying to reduce the distance from the empirical distribution $\hat{\mathbb{P}}_N$. It is instructive to note that the distance matric $d(\cdot, \cdot)$ plays an important role to determine the solution of distribution $\mathbb{Q}$.

## 2.1 Learning via Generation

We introduce the Wasserstein distance $d_W(\cdot, \cdot)$ to describe the distance between distributions $\hat{\mathbb{P}}_N$ and $\mathbb{Q}$. Wasserstein distance measures the distance between two distributions as the minimum variation required to transition from one distribution to another. We define the Wasserstein distance as follows,

$$d_W(\mathbb{Q}_1, \mathbb{Q}_2) \triangleq \min_{\Pi \in \mathcal{M}(\Pi)} \left\{ \int_{\mathcal{Z} \times \mathcal{Z}} s((\boldsymbol{x}_1, y_1), (\boldsymbol{x}_2, y_2)) \Pi\left(\mathrm{d}(\boldsymbol{x}_1, y_1), \mathrm{d}(\boldsymbol{x}_2, y_2)\right) \right\}, \tag{6}$$

where $\Pi$ is defined as the joint distribution of $(\boldsymbol{x}_1, y_1)$ and $(\boldsymbol{x}_2, y_2)$ with marginals $\mathbb{Q}_1$ and $\mathbb{Q}_2$ respectively, $\mathcal{M}(\Pi)$ represents the space of all probability of $\Pi$, and $s$ is a metric measuring the cost of moving $(\boldsymbol{x}_1, y_1)$ to $(\boldsymbol{x}_2, y_2)$. The calculation of the Wasserstein distance is not so straightforward because of the need to find an optimal joint distribution $\Pi$ that minimizes the integral value. According to [29], the Wasserstein distance can be further calculated as follows:

$$d_W(\mathbb{Q}_1, \mathbb{Q}_2) = \sup_{f \in \mathcal{F}} \left\{ \int_{\mathcal{Z}} f(\boldsymbol{x}, y) \mathbb{Q}_1(\mathrm{d}(\boldsymbol{x}, y)) - \int_{\mathcal{Z}} f(\boldsymbol{x}, y) \mathbb{Q}_2(\mathrm{d}(\boldsymbol{x}, y)) \right\}, \tag{7}$$

where $\mathcal{F}$ donates the space of all Lipschitz function with $|f(\boldsymbol{t}) - f(\boldsymbol{t}')| \leq \|\boldsymbol{t} - \boldsymbol{t}'\|$ for all $\boldsymbol{t}$ and $\boldsymbol{t}' \in \mathcal{T}$. Eq. (7) replaces the optimal joint distribution $\Pi$ involved in the Eq. (6) by finding a specific function $f$ in a function set $\mathcal{F}$. We can get Eq. (7) as the distance matric into the Eq. (5).

$$\sup_{\mathbb{Q}} \int_{\mathcal{X}} \ell_\theta(\boldsymbol{x}, y) \mathbb{Q}\left(\mathrm{d}(\boldsymbol{x}, y)\right) - \lambda \cdot \sup_{f \in \mathcal{F}} \left\{ \int_{\mathcal{X}} f(\boldsymbol{x}, y) \hat{\mathbb{P}}_N(\mathrm{d}(\boldsymbol{x}, y)) - \int_{\mathcal{X}} f(\boldsymbol{x}, y) \mathbb{Q}(\mathrm{d}(\boldsymbol{x}, y)) \right\}$$

$$= \sup_{\mathbb{Q}_i} \frac{1}{N} \sum_{i=1}^{N} \int_{\mathcal{X}} \ell_\theta(\boldsymbol{x}, y) \mathbb{Q}_i\left(\mathrm{d}(\boldsymbol{x}, y)\right) - \lambda \cdot \sup_{f \in \mathcal{F}} \left\{ \frac{1}{N} \sum_{i=1}^{N} \left[ f(\boldsymbol{x}_i, y_i) - \int_{\mathcal{X}} f(\boldsymbol{x}, y) \mathbb{Q}_i(\mathrm{d}(\boldsymbol{x}, y)) \right] \right\} \tag{8}$$

$$= \frac{1}{N} \sum_{i=1}^{N} \sup_{(\boldsymbol{x}, y) \sim \mathbb{Q}} \left\{ \ell_\theta(\boldsymbol{x}, y) - \lambda \cdot [\hat{f}(\boldsymbol{x}_i, y_i) - \hat{f}(\boldsymbol{x}, y)] \right\},$$

where $\hat{f} = \operatorname{argmax}_{f \in \mathcal{F}} \frac{1}{N} \sum_{i=1}^{N} f(\boldsymbol{x}_i, y_i) - \frac{1}{N} \sum_{i=1}^{N} \int_{\mathcal{X}} f(\boldsymbol{x}, y) \mathbb{Q}_i(\mathrm{d}(\boldsymbol{x}, y))$ is the optimized function to describe the Wasserstein distance between the desirable distribution $\mathbb{Q}$ and the empirical distribution $\hat{\mathbb{P}}_N$, $\mathbb{Q}_i$ is the conditional distribution of $(\boldsymbol{x}, y)$ given $(\boldsymbol{x}_i, y_i)$, The joint distribution $\Pi$ of $(\boldsymbol{x}_i, y_i)$ and $(\boldsymbol{x}, y)$ with marginals $\hat{\mathbb{P}}_N$ and $\mathbb{Q}$ respectively (see Eq. (3)), can be written as $\Pi = \frac{1}{N} \sum_{i=1}^{N} \delta_{(\boldsymbol{x}_i, y_i)} \otimes \mathbb{Q}_i$. According to the law of total probability, we can factorize $\mathbb{Q}$ as the first line of Eq. (8). Eq. (8) bridges the training sample and the distribution $\mathbb{Q}$, and $\mathbb{Q}$ is thus defined by:

$$\mathbb{Q} = \operatorname*{argmax}_{\mathbb{Q}} \frac{1}{N} \sum_{i=1}^{N} \left\{ \ell_\theta(\boldsymbol{x}, y) - \lambda \cdot [\hat{f}(\boldsymbol{x}_i, y_i) - \hat{f}(\boldsymbol{x}, y)] \right\}$$

$$= \operatorname*{argmax}_{\mathbb{Q}} \mathbb{E}_{\mathbb{Q}}[\ell_\theta(\boldsymbol{x}, y) + \lambda \cdot \hat{f}(\boldsymbol{x}, y)] - \lambda \cdot \mathbb{E}_{\hat{\mathbb{P}}_N}[\hat{f}(\boldsymbol{x}, y)] \tag{9}$$

$$= \operatorname*{argmax}_{\mathbb{Q}} \mathbb{E}_{\mathbb{Q}}[\ell_\theta(\boldsymbol{x}, y) + \lambda \cdot \hat{f}(\boldsymbol{x}, y)].$$

A neural network $G(\boldsymbol{z})$ can be employed to approximate the distribution $\mathbb{Q}$, and thus Eq. (9) can be rewritten as the maximization of $\mathbb{E}_{\mathcal{Z}}[\ell_\theta(G(\boldsymbol{z})) + \lambda \cdot \hat{f}(G(\boldsymbol{z}))]$. According to Eq. (8), we need to solve an optimal $\hat{f}$ to calculate the Wasserstein distance. We also adopt a neural network $D$ for help, and propose to maximize $\frac{1}{N} \sum_{i=1}^{N} D(\boldsymbol{x}_i, y_i) - \frac{1}{N} \sum_{i=1}^{N} \int_{\mathcal{X}} D(\boldsymbol{x}, y)) \mathbb{Q}_i(\mathrm{d}(\boldsymbol{x}, y))$. Finally, we can learn the classifier over the bad data by considering the worst distribution case through the following objective function

$$\min_{G} \max_{D, C} U(C, G, D) = \lambda \big( \mathbb{E}_{\hat{\mathbb{P}}_N}[D(\boldsymbol{x}, y)] - \mathbb{E}_{\mathbb{Q}}[D(\boldsymbol{x}, y)] \big) - \mathbb{E}_{\mathbb{Q}}[\ell(C(\boldsymbol{x}), y)]. \tag{10}$$

Recall that $\lambda$ is a Lagrangian multiplier in Eq. (4). According to the analysis in the Lagrange multiplier method, the larger $\lambda$ corresponds to a smaller epsilon, which implies that the $\mathbb{Q}$ distribution is closer to the $\hat{\mathbb{P}}_N$ distribution. Conversely, a smaller $\lambda$ will allow a larger distribution distance $\epsilon$, which allows the $\mathbb{Q}$ distribution to be explored over a sufficiently large range. Following we provide intuitive explanation why the worst-case optimization works. Minimizing the loss of the worst-case distribution $\mathbb{Q}$ implies an optimization over all distributions within the ball of an appropriate radius $\epsilon$ (see Eq. (1)), which could also include the unknown real distribution $\mathbb{P}_N$. Though the worst-case $\mathbb{Q}$ may not be exactly the real $\mathbb{P}_N$, *the classifier (i.e. $\theta$) must have fitted $\mathbb{P}_N$ better than (or equivalently with) $\mathbb{Q}$*, as the classification error over $\mathbb{Q}$ is the worst. In iterations, the worst-case $\mathbb{Q}$ will be dynamically determined by the classifier, and the classifier will fit the real $\mathbb{P}_N$ increasingly better in an implicit way.

**Difference from GANs.** Though we also introduce a generator and investigate an adversarial game, our model has several differences from existing GAN models. Compared with WGAN [3], besides the critical network $D$, our generative network $G$ further plays against the classification network $C$. There are some three-way GAN models, such as Triple GAN [27], Triangle GAN [10] (actually 4 players), and ALI [9]. These models have two opposite generation models, $C : \boldsymbol{x} \to y$ and $G : y \to \boldsymbol{x}$. At first, we have a different motivation to establish the adversarial games, compared with these existing methods. Triple GAN and Triangle GAN are dedicated to a semi-supervised learning, while ALI are dedicated to improving the training of $D$ network by learning a set of opposite mappings from $y$ to $x$. In contrast, we aim to learn a classifier that can deal with bad data, and our adversarial model is to provide an appropriate measure between the worst case distribution and the empirical data distribution. In addition, existing methods for implementing the two sets of opposite mappings shares the same goal, that is to deceive $D$ network, and there is no explicit relationship between the two generators (*i.e. $G$ and $C$*). However, our generator deceives not only the discriminator, but also the classifier. That is to say, our two opposite mappings are directly competitive with each other.

## 3 Theoretical Analysis

In the proposed method, the classifier is optimized on a learned distribution $\mathbb{Q}$. $\mathbb{Q}$ represents the worst-case distribution within a certain range, which is a key point of the entire algorithm. In this section, we provide a formal technical analysis of the convergence of the three networks to better understand the relationship between the three networks. Next, by analyzing the difference between the experimental distribution $\hat{\mathbb{P}}_N$ and the learned distribution $\mathbb{Q}$, we prove that our algorithm can be regarded as a data-dependent gradient regularization, which provides a reason for improvement of the generalization ability provided by the proposed algorithm. See the supplementary material for proofs.

**Influence of optimal $D$ and $G$.** In the framework defined by Eq. (10), the critical network $D$ attempts to fit the desirable function $\hat{f} = \operatorname{argmax}_{f \in \mathcal{F}} \frac{1}{N} \sum_{i=1}^{N} f(\boldsymbol{x}_i, y_i) - \frac{1}{N} \sum_{i=1}^{N} \int_{\mathcal{X}} f(\boldsymbol{x}, y) \mathbb{Q}_i(\mathrm{d}(\boldsymbol{x}, y))$ which represents the Wasserstein distance between two distribution. As a result, the optimal critical network is expected to describe the Wasserstein distance perfectly, which is $\mathbb{E}_{\hat{\mathbb{P}}_N}[D^*(\boldsymbol{x}, y)] - \mathbb{E}_{\mathbb{Q}}[D^*(\boldsymbol{x}, y)] = d_W(\hat{\mathbb{P}}_N, \mathbb{Q})$.

Next we analyze the target distribution of the generator $G$. As described in Eq. (10), the generator aims to maximize loss $\lambda E_{\mathbb{Q}}[D(\boldsymbol{x}, y)] + \mathbb{E}_{\mathbb{Q}}[\ell(C(\boldsymbol{x}), y)]$. In Theorem 1 we summarize the equilibrium distribution obtained by $G$ which is also determined by $\lambda$.

**Theorem 1.** *With the optimal critical network $D$ and the classifier $C$ fixed, the optimization of generator $G$ is equivalent to minimize $\lambda \cdot d_W(\hat{\mathbb{P}}_N, \mathbb{Q}) - D_{KL}(\mathbb{Q}||\mathbb{P}_c)$.*

Theorem 1 suggests that the distribution $\mathbb{Q}$ will be optimized to be as far as possible away from the distribution $\mathbb{P}_c$ while towards the distribution $\hat{\mathbb{P}}_N$. Distribution $\mathbb{Q}$ will fit a worse distribution for $C$ iterative and enforce $C$ to be optimized over the whole distribution set $\mathbb{B}_\epsilon$.

**Data-dependent regularization.** Traditional methods used to optimize the classification network $C$ over the empirical distribution $\hat{\mathbb{P}}_N$. From the perspective of network complexity, some regularization are often introduced to improve the generalization capabilities of the network, such as weight decay and dropout. We theoretically show that by setting the distance metric as Wasserstein distance, we will derive a data-dependent gradient regularization.

**Theorem 2.** *Consider $\boldsymbol{x}$ as the input samples of classifier $C$, and the distribution $\mathbb{Q} \in \mathbb{B}_\epsilon(\hat{\mathbb{P}}_N)$ lays in a Wasserstein ball centered at $\hat{\mathbb{P}}_N$ with radius $\epsilon$. Then for any $\epsilon \geq 0$ and $\alpha \geq 1 + \beta$, we have*

$$\epsilon\|\nabla_{\boldsymbol{z}}\ell(\boldsymbol{z})\|_{\hat{\mathbb{P}}_N}^{\alpha_*} - \epsilon^{\beta+1}\|h(\boldsymbol{z})\|_{\hat{\mathbb{P}}_N}^{\frac{\alpha}{\alpha-\beta-1}} \leq \mathbb{E}_{\mathbb{Q}}(\ell(\boldsymbol{z})) - \mathbb{E}_{\hat{\mathbb{P}}_N}(\ell(\boldsymbol{z})) \leq \epsilon\|\nabla_{\boldsymbol{z}}\ell(\boldsymbol{z})\|_{\hat{\mathbb{P}}_N}^{\alpha_*} + \epsilon^{\beta+1}\|h(\boldsymbol{z})\|_{\hat{\mathbb{P}}_N}^{\alpha_*}, \quad (11)$$

*where $\|f(\boldsymbol{z})\|_{\hat{\mathbb{P}}_N}^{\alpha} \triangleq \left(\frac{1}{N}\sum_{i=1,\boldsymbol{z}\sim\hat{\mathbb{P}}_N}^{N}(\|f(\boldsymbol{z}_i)\|^{\alpha})\right)^{1/\alpha}$, $\alpha_* = \frac{\alpha}{\alpha-1}$, and $h(\boldsymbol{z})$ is a function and $\beta \in (0,1]$ a constant which satisfy $\|\nabla_{\boldsymbol{z}}\ell(\boldsymbol{z}_1) - \nabla_{\boldsymbol{z}}\ell(\boldsymbol{z}_2)\| \leq h(\boldsymbol{z}_2) \cdot \|\boldsymbol{z}_1 - \boldsymbol{z}_2\|^{\beta}$ for any $\boldsymbol{z} = (\boldsymbol{x}, y) \in \mathcal{Z}$.*

It shows that optimization over the worst-case distribution $\mathbb{Q}$ can be roughly interpreted as a data-driven gradient regularization. By minimizing the loss function $\ell(\boldsymbol{z})$ over $\mathbb{Q}$, the gradient of the loss function with respect to the empirical samples $\nabla_{\boldsymbol{x}}\ell(C(\boldsymbol{x}_i,\theta),y)$ will also be optimized. Furthermore, gradient penalties applied over the empirical sample also lead the classifier $C$ to react more gently to changes in the sample, which provides another perspective on the effectiveness of our algorithm.

**Performance guarantees.** In this part, we analyze the generalization capabilities of the classifiers obtained by the proposed method. The generalization ability of the classifier is often described as the bias in the performance of the network between training samples $\boldsymbol{x} \in \mathbb{Q}$ and the new sample $\tilde{\boldsymbol{x}}$, and the smaller the bias corresponds to better generalization. In the following theorem, we propose a bound for the predictive performance of the classifier on the new sample.

**Theorem 3.** *For any $0 < \delta < 1$, with probability at least $1 - \delta$ with respect to the sampling,*

$$\mathbb{E}(\ell(C(\boldsymbol{x},\boldsymbol{\theta}),y)) \leq \mathbb{E}_{\mathbb{Q}}(\ell(C(\boldsymbol{x},\boldsymbol{\theta}),y)) + \frac{12\sqrt{R}}{n}(\log\frac{n}{3\sqrt{R}} + 1) + \sqrt{\frac{8\log(2/\delta)}{N}}, \quad (12)$$

*and for any $\zeta > \frac{12\sqrt{R}}{n}(\log\frac{n}{3\sqrt{R}} + 1) + \sqrt{\frac{8\log(2/\delta)}{N}}$, we have*

$$P\left(\ell(C(\boldsymbol{x},\boldsymbol{\theta}),y) \geq \mathbb{E}_{\mathbb{Q}}(\ell(C(\boldsymbol{x},\boldsymbol{\theta}),y)) + \zeta\right) \leq \frac{\mathbb{E}_{\mathbb{Q}}(\ell(C(\boldsymbol{x},\boldsymbol{\theta}),y)) + \frac{12\sqrt{R}}{n}(\log\frac{n}{3\sqrt{R}} + 1) + \sqrt{\frac{8\log(2/\delta)}{N}}}{\mathbb{E}_{\mathbb{Q}}(\ell(C(\boldsymbol{x},\boldsymbol{\theta}),y)) + \zeta}.$$
$$(13)$$

*where $R$ is only related to the architecture of the neural network.*

We leave the detailed definition of $R$ in the appendix. Theorem 3 provides a bound over the error of the classifier on new samples, which contains two probability measures. Wherein, Eq. (12) indicates that the error of the classifier on new samples does not exceed $\frac{12\sqrt{R}}{n}(\log\frac{n}{3\sqrt{R}} + 1)$ at a probability of at least $1 - \delta$, where $N$ is the number of training samples. Eq. (13) provides an upper bound for the probability that the classifier's error on the new sample exceeds $\frac{12\sqrt{R}}{n}(\log\frac{n}{3\sqrt{R}} + 1)$. From Eq. (12), we find that the bound of the classifier error is related to the number of training samples $N$, and the larger the number of samples, the smaller the error. In our training framework, the generator $G$ is responsible for producing training samples, and $G$ is also updated along with the classifier $C$. Since that, the training samples far exceed that of the traditional algorithms, which will reduce the error of the classifier on the new data, and the generalization ability of the classifier will be improved.

## 4 Extension to Other Distance Measure

The proposed model can be extended to a standard GAN based game by investigating Jensen-Shannon (JS) divergence between distributions. The critical network $D$ is designed to fit the desirable function $\hat{f}(\cdot)$ and calculate Wasserstein distance between distribution $\mathbb{Q}$ and the empirical distribution $\hat{\mathbb{P}}_N$. In standard GAN, the discriminator acts as a classifier and attempts to distinguish fake samples generated by $G$ from real samples. The objective function of the discriminator can be written as $\mathbb{E}_{\hat{\mathbb{P}}_N}[\log(D(\boldsymbol{x},y))] + \mathbb{E}_{\mathbb{Q}}[1 - \log(D(\boldsymbol{x},y))]$. By simply replacing the objective function of the network $D$, we can formulate the three-players game based on a standard GAN as follows,

$$\min_{C,D}\max_{G} U(C,G,D) = \lambda(\mathbb{E}_{\hat{\mathbb{P}}_N}[\log(D(\boldsymbol{x},y))] + \mathbb{E}_{\mathbb{Q}}[1 - \log(D(\boldsymbol{x},y))]) + \mathbb{E}_{\mathbb{Q}}[\ell(C(\boldsymbol{x},\boldsymbol{\theta}),y)] \quad (14)$$

As described in Eq. (14), the generator G whose responsibility is to fit the objective distribution $\mathbb{Q}$ and tries to fool both the discriminator $D$ and the classifier $C$. The confrontation with the discriminator $D$ leads the generator $G$ to produce samples that are as close as possible to the true distribution. At the same time, these samples also make the performance of the classifier $C$ worse.

Now we consider the standard GAN based framework and analyze the optimal discriminator and generator. The discriminator $D$ is optimized by $\mathbb{E}_{\hat{\mathbb{P}}_N}[\log(D(\boldsymbol{x}, y))] + \mathbb{E}_{\mathbb{Q}}[1 - \log(D(\boldsymbol{x}, y))]$, which can be considered as distinguishing generated samples $(\boldsymbol{x}, y) \sim \mathbb{Q}$ from the true sample $(\boldsymbol{x}, y) \sim \hat{\mathbb{P}}_N$. Following the analysis proposed in GAN [14], the optimal distribution $D$ will balance between the true distribution $\hat{\mathbb{P}}_N$ and the learned distribution $\mathbb{Q}$.

**Theorem 4.** *For the generator $G$ and classifier $C$ fixed, the optimal discriminator $D$ is*

$$D_{G,C}^*(\boldsymbol{x}, y) = \frac{p_{data}(\boldsymbol{x}, y)}{p_{data}(\boldsymbol{x}, y) + p_g(\boldsymbol{x}, y)}, \tag{15}$$

*where $p_g(\boldsymbol{x})$ is the distribution generated by $G$.*

With the optimal discriminator $D$ fixed, we can reformulate the objective function by replacing $D(\boldsymbol{x}, y)$ in Eq. (14) according to Theorem 4. By doing so, we show that the optimal generator $G$ will also balance between the empirical distribution $\hat{\mathbb{P}}_N$ and the distribution $\mathbb{Q}$ which is represented by classifier $C$, as summarized in the following theorem.

**Theorem 5.** *With the optimal discriminator $D$ and the classifier $C$ fixed, the optimization of generator $G$ is equivalent to $-\log 4 + 2JSD(\hat{\mathbb{P}}_N||\mathbb{Q}) - 1/\lambda \cdot D_{KL}(\mathbb{Q}||\mathbb{P}_c)$.*

**Justification of the standard GAN based framework.** The distribution $\mathbb{Q}$ obtained by Eq. (10) is a straightforward result of Eq. (3) and satisfies two conditions. The first one is the distance between distributions $\hat{\mathbb{P}}_N$ and $\mathbb{Q}$ is less than a constant $\epsilon$, and the second one is to make the classification loss as bad as possible. As Eq. (14) is obtained by simply replacing the critical loss with a discriminator loss $\mathbb{E}_{\hat{\mathbb{P}}_N}[\log(D(\boldsymbol{x}, y))] + \mathbb{E}_{\mathbb{Q}}[1 - \log(D(\boldsymbol{x}, y))]$, whether or not the distribution $\mathbb{Q}$ obtained in Eq. (14) satisfies such conditions of Eq. (3) cannot be easily justified. The distribution $\mathbb{Q}$ obtained by Eq. (14) is optimized according to the minimization of $-\log 4 + 2JSD(\hat{\mathbb{P}}_N||\mathbb{Q}) - \lambda \cdot D_{KL}(\mathbb{Q}||\mathbb{P}_c)$. By comparing Theorem 5 and Theorem 1, with an ignorance of the constant term $-\log 4$, we can find that the major difference between the equilibrium distributions of Eq. (10) and Eq. (14) is the choice of distance metric , *i.e.*, the Wasserstein distance $d_W(\hat{\mathbb{P}}_N||\mathbb{Q})$ for Eq. (10) and the JS divergence $JSD(\hat{\mathbb{P}}_N||\mathbb{Q})$ for Eq. (14). We now build a relationship between the initial objective and Eq. (14) and draw a conclusion that the loss function defined by Eq. (14) can be viewed as the JS divergence version of Eq. (3). The JS divergence based objective function shares the same training procedure with the Wasserstein distance one. Our proposed algorithm is summarized in Algorithm 1 in Appendix.

## 5 Experiments

In this section, we evaluate our methods on three kinds of bad data environments: $(i)$ long-tailed training set classification on the MNIST [25], FMNIST [42], and CIFAR-10 [23] datasets; $(ii)$ classification of distorted test set on the CIFAR-10 and SVHN [33] datasets; and $(iii)$ reduced training set generation task on the FMNIST and CIFAR-10 datasets. We resize images in the MNIST and FMNIST datasets to $32 \times 32$ for convenience. Moreover, we use a conditional version of WGAN-GP [15] on all datasets except the CIFAR-10 datasets on which we use the 32 resolution version of BigGAN [4] instead. The classifier implemented on the MNIST and FMNIST has comparable architecture to Triple GAN [27], and we use VGG-16 [38] and ResNet-101 [18] on the CIFAR-10 and SVHN datasets. We implement our experiments based on PyTorch. For generator and discriminator we use a learning rate of 0.0002, while 0.02 is for the classifier, the learning rate decay is deployed, and the optimizer is Adam. Experiments are conducted on 4 NVIDIA 1080Ti GPUs.

### 5.1 Classification results

For the long tail experiment, we transform the original balanced training set according to an exponential function $n = n_i \times \mu^i$, where constant $\mu \in (0, 1)$, and $n_i$ is the original number of sample of category $i$. Under this setting, we follow definition in [5] and define the imbalance factor as the number of training samples in the largest class divided by the smallest one. We compare the proposed method with two the state-of-the-art algorithms which are Class-Balanced [5] and DOS [1] respectively. In experiments on noisy test sets, we introduce a certain intensity of Gaussian noise or salt-and-pepper noise into 70% of the test samples to produce noisy test sets. Moreover, in experiment of noisy test sets, GAN-based methods use ResNet-101 as classifier and MixQualNet [6]

Table 1: Accuracy (%) on long-tailed datasets with various imbalance factors.

| Method | MNIST | | | FMNIST | | | CIFAR | | |
|---|---|---|---|---|---|---|---|---|---|
| Imbalanced | 100 | 20 | 1 | 100 | 20 | 1 | 20 | 10 | 1 |
| Classifier | 89.77 | 94.62 | 99.35 | 79.89 | 87.33 | **93.46** | 81.51 | 85.57 | **93.04** |
| DAGAN [2] | 89.92 | 95.71 | 99.23 | 77.63 | 86.30 | 92.87 | 70.34 | 77.82 | 91.66 |
| Triple GAN [27] | 90.07 | 95.19 | 99.31 | 78.92 | 87.83 | 92.55 | 80.40 | 83.06 | 92.81 |
| $\Delta$-GAN [10] | 90.25 | 95.60 | 99.28 | 78.85 | 87.62 | 93.06 | 79.99 | 85.47 | 92.87 |
| Ours | **92.67** | **96.23** | **99.42** | **83.06** | **89.03** | 93.24 | **81.94** | **86.86** | 93.01 |
| C-B Loss [5] | **92.90** | 96.74 | **99.38** | 83.77 | 89.97 | 93.43 | 84.36 | 87.49 | 93.64 |
| DOS [1] | 90.82 | **97.20** | 99.07 | 82.74 | 89.34 | 93.18 | 81.72 | 86.55 | 92.83 |

Table 2: Accuracy (%) on the clean train sets and distorted test sets of CIFAR-10 and SVHN.

| Method | CIFAR-10 | | | | SVHN | | | |
|---|---|---|---|---|---|---|---|---|
| | $G(0.2)$ | $G(0.3)$ | $S(0.02)$ | Normal | $G(0.5)$ | $G(0.7)$ | $S(0.1)$ | Normal |
| VGG16 [38] | 82.98 | 63.09 | 63.87 | 91.86 | 94.84 | 94.09 | 93.70 | 97.63 |
| ResNet [18] | 84.41 | 64.15 | 64.53 | **93.04** | 95.25 | 94.52 | 94.23 | 98.17 |
| DAGAN [2] | 80.67 | 61.19 | 61.38 | 91.66 | 94.64 | 94.45 | 93.91 | 97.82 |
| Triple GAN [27] | 84.57 | 63.76 | 64.51 | 92.81 | 95.21 | 94.60 | 94.09 | 97.85 |
| $\Delta$-GAN [10] | 84.46 | 64.28 | 64.59 | 92.87 | 95.42 | 94.37 | 94.14 | 97.76 |
| Ours-VGG | 85.02 | 64.43 | 65.26 | 92.32 | 95.08 | 94.55 | 94.04 | 97.69 |
| Ours-ResNet | **85.87** | **65.43** | **66.80** | 93.01 | **95.58** | **95.02** | **94.67** | **98.33** |
| MixQualNet [6] | **86.56** | **65.70** | 66.71 | 89.62 | 95.48 | **95.27** | 94.21 | 96.80 |
| DCTNet [34] | 85.42 | 65.68 | **69.13** | **90.93** | **95.55** | 95.10 | **94.88** | **97.41** |

and DCTNet [34] get their results based on VGG16. We also investigated the performance of our approach when using VGG and ResNet-101 as classifier respectively and reported results in Table 2.

Table 1 and Table 2 report results obtained on long-tailed and noisy datasets respectively. Note that imbalance factor (IF) of 1 means that the class is balance. In Table 2, $G$ and $S$ represent Gaussian noise and salt-and-pepper noise respectively, and the number represents standard deviation in $G$ and noise rate in $P$. It is obviously that the performance of the classifier drops significantly as the imbalance factor increases. We implement DAGAN to achieve data augmentation and train classifier with these data. However, the improvement is slight on the MNIST dataset, and the performance even drops on more complicated datasets such as FMNIST and CIFAR-10. It indicates that samples generated by GAN help less for classifier. Triple GAN and Triangle GAN ($\Delta$-GAN) show more improvement than that of DAGAN but are not stable enough. This phenomenon can be interpreted by their architectures, where the generator pleases the classifier rather than playing against it. The proposed method outperforms the other GAN-based methods and achieves the best results on most conditions. We conclude the reason in following two points. First, generators in existing methods tend to fit the empirical distribution. Given a bad training set, their generated data could be worse. Second, these generators often produce "easy" samples by cooperating with the classifier, and a nearly duplicate copy of the given bad training data could be sufficient but will be useless to estimate the real data distribution. In contrast, our generator plays against the classifier and the capability of the classifier can be largely enhanced over a distribution ball. Moreover, most GAN-based methods failed to provide performance improvement on the normal datasets (IF of 1 or 'Normal'), but our method even outperforms classifier on the MNIST and SVHN datasets. Those results may because that though training data are clean, there still probably exists a *subtle* gap between distributions of training and test data. Moreover, the generator could conduct 'data augmentation' for the classifier. We also compared our method with state-of-the-art algorithms. Noting that these algorithms are designed for a specific type of data defect, the proposed method also achieves comparable results with these algorithms in each situation.

**Comparison with data augmentation methods.** Considering the generator trained by the proposed algorithm as a learned data augmenter, the proposed method can be viewed as a data augmentation method. In this part we evaluate two common data augmentation methods with the proposed method on the CIFAR-10 dataset. The first method is a combination of reg-

Table 3: Accuracy (%) on CIFAR-10.

| Method | IF = 10 | G(0.2) | Reduced |
|---|---|---|---|
| Combination | 85.63 | 85.25 | 83.59 |
| Mixup [44] | 86.04 | 85.66 | 83.91 |
| Ours | **86.86** | **85.87** | **84.60** |

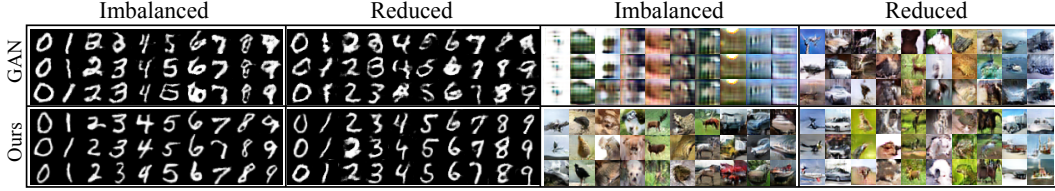

Figure 1: Generated images obtained by GAN and our method on imbalanced and reduced dataset.

ular data augmentation methods, which are randomly
clip, horizon flip, and rotation, and we show its results in the first line of Table 3. The second data augmentation method is Mixup [44]. Table 3 shows that our method outperforms both two comparison methods. Mixup also provides reasonable improvement but is not as outstanding as it on regular datasets.

## 5.2 Analysis

**Generation results** We compare the quality of images generated by our generator with GAN and Triple GAN on the MNIST and CIFAR-10 datasets with imbalance or reduced training set. The imbalance factor used here is 10, and we obtain reduced training set by randomly selecting 30% of samples from the training set. We compared the quality of images generated by our method with standard GAN in Figure 1, and FID scores are reported in Table 4. We obtain the feature used for calculating FID from the specific layer of the pre-trained inception model. FIDs are calculated with 10,000 samples randomly chosen from training dataset and 10,000 generated samples. On imbalanced training sets, standard GAN is failed to generate high-quality images for classes with fewer images especially in the CIFAR-10 dataset, while our method generates images with satisfied quality. In the MNIST dataset, the proposed method achieves a higher FID score than Triple GAN, but our classifier obtains a higher accuracy as showed in Table 1. It indicates that the generator in our method does not generate images with the best quality but do generate more helpful images for the classifier. On the reduced training set, our method outperforms other algorithms. As the true images used to calculate FID scores are sampled from the whole training set instead of the reduced one, the FID represents how close the generated distribution is to the true distribution, instead of the empirical one. Minimizing the worst-case expected loss implies an optimization over all distributions in the $\epsilon$-ball where the real data distribution is also expected to be included.

| | Table 4: FIDs with different distributions. | | | |
|---|---|---|---|---|

| Method | MNIST | | CIFAR-10 | |
|---|---|---|---|---|
| | IF=10 | Reduced | IF=10 | Reduced |
| GAN | 33.49 | 31.06 | None | 37.96 |
| Triple | **27.24** | 26.30 | 26.22 | 22.57 |
| Ours | 27.60 | **26.02** | **26.10** | **22.63** |

| | Table 5: Ablation study results. | | |
|---|---|---|---|

| Method | CNN | DAGAN | Ours |
|---|---|---|---|
| BN+WD | 83.19 | 83.08 | **84.60** |
| No BN | 75.14 | 71.92 | **78.01** |
| No WD | 77.01 | 75.38 | **78.71** |
| Neither | 71.85 | 71.63 | **75.42** |

**Ablation study** Weight decay (WD) [24] and Batch Normalization (BN) [21] are considered as common methods to increase the robustness of the network. To illustrate the effectiveness of our method, we implement experiments on four classifier architectures: (i) classifier with BN and WD, (ii) classifier with BN (without WD), (iii) classifier with WD (without BN), and (iv) classifier only (without BN nor WD). Results in Table 5 are obtained on reduced CIFAR-10 dataset which contains 20% samples of original training set, and the classifier is set as ResNet-101. Table 5 shows that the proposed method not only outperforms classifier and DAGAN but enjoys a smaller accuracy rate drop when the network structure changes. It proves that our method can provide more challenging images for classifier and play the same role as these generalization algorithms.

**Hyper-parameter analysis** As shown in Eq. (2), a large $\epsilon$ leads to a set $\mathbb{B}_\epsilon$ of huge capacity, which could be flooded with distributions that are far away from both the empirical distribution $\hat{\mathbb{P}}_N$ and the real data distribution $\mathbb{P}_N$. It is therefore reasonable to set $\epsilon$ within an appropriate range, as what we usually do with hyper-parameters in machine learning. For a better understanding of the role of $\epsilon, \lambda$ proposed in Eq. (2) and Eq. (9) respectively, we use the long-tailed CIFAR-10 dataset with imbalance factor of 10 to show the accuracy of the proposed method in Figure 2 (d). The search for

hyper-parameter is $\epsilon = \frac{1}{\lambda} \in \{0.01, 0.1, 0.3, 0.5, 1.0, 2.0\}$. We have the following observations that smaller value of $\epsilon$ will make the accuracy of the classifier closer to the general classifier, too large value of $\epsilon$ will drop the accuracy of classifier dramatically, and the best $\epsilon$ is 0.3 on this dataset. These results can be explained by the theoretical analysis section. According to Theorem 1 and 5, a large $\lambda$ will make the distribution $\mathbb{Q}$ close to the empirical distribution $\hat{\mathbb{P}}_N$ which makes the performance of classifier to be similar to the general classifier, and a small $\lambda$ will lead the distribution $\mathbb{Q}$ only to cheat the classifier while ignoring the quality of the generated image.

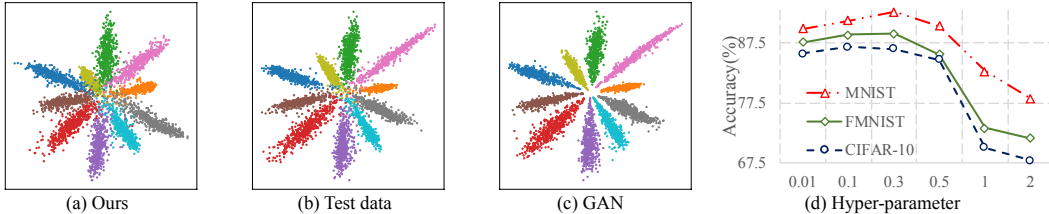

| (a) Ours | (b) Test data | (c) GAN | (d) Hyper-parameter |

Figure 2: (a-c) Feature visualization for different samples. (d) Hyper-parameter analysis.

**Visualization** In Figure 2 we visualize the second last layer features of images sampled from empirical distribution ($\boldsymbol{x} \sim \hat{\mathbb{P}}_N$), generated by GAN, and generated by the proposed method ($\boldsymbol{x} \sim \mathbb{Q}$) on the reduced MNIST dataset (30% of traning set). We obtain features by forwarding images to a classifier pre-trained on the reduced training set, and features with a specific category in each figure are represented in the same color. Figure 2 demonstrates that features from GAN shows less diversity and can be easily distinguished (shown in (c)), features from $\mathbb{P}_{test}$ tend to confuse the classifier (shown in (b)), and features from our generator are more challenging for the classifier (shown in (a)). This shows the effectiveness of the proposed method in learning the worst-case distribution.

## 6 Related Work

To obtain robustness, a straightforward way is to train deep networks with the expected perturbations [41]. A mixture of the expert classifiers that is trained by various types of image perturbation is proposed [6] and shows more robustness than previous single model methods [45]. To resolve the heavy parameters brought by the ensemble of many networks, additional layers [43] was introduced to the network. They act as undistorted layers and improve the robustness of the network by reconstructing input images. For long-tailed imbalanced training data, re-sampling and cost-sensitive methods are two major strategies. Re-Sampling includes over-sampling which duplicates samples in rare classes and under-sampling, which deletes samples from common classes. Over-sampling [16, 28] is limited by the repeated samples, which leads the network to overfit, while under-sampling [8] suffers from the information loss caused by samples deleting. Cost-sensitive methods consider samples with different weight when calculating the loss function. There are methods assigning weights according to the class frequency [19, 32] and assigning weights to the samples based on how difficult it is to be resolved by the network [29, 7], which is somewhat similar to the proposed method. Reduced training data is also a challenging task in classification. Some data augmentation algorithms are proposed to relieve the shortage of training data, such as DAGAN [2] and Smart Augmentation [26]. DAGAN also introduces GAN [14] to generate samples and trains classifier. But we are different from them in that they did not consider the classifier in the process of generating samples like what we do. In addition, adversarial data augmentation usually aims to create adversarial copies of training data by adding perturbations. In contrast, we generate new sample from a distribution. Moreover, the amount of perturbation (pixels for image) is often constrained in classical methods, while we focus on a distribution ball with a radius bound.

## 7 Conclusion

We propose a new adversarial classification algorithm that improves the performance of the classifier when a gap exists between the unknown true distribution and known empirical distribution. By dynamically interacting with classifiers and known data distributions, a worst case distribution is learned to help the training progress of classifier, which is in contrast to existing robust algorithms for one specific data defect. Both theoretical analysis and experimental results show that the proposed method can effectively improve the generalization ability of classifiers on bad data sets.

## Acknowledgment

We thank anonymous area chair and reviewers for their helpful comments. This research was supported in part by National Natural Science Foundation of China under Grant No. 61876007 and 61872012, and Australian Research Council Grant DE-180101438.

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
