[Supplementary Material · nips19_3262_supp.pdf]

# Supplementary Material of
# Learning from Crap Data via Generation

## 1 A Proofs

2 In this section, we denote $\boldsymbol{z} = (\boldsymbol{x}, y)$ and $\boldsymbol{z} \in \mathcal{Z}$ for convenience.

### 3 A.1 Proof of Theorem 1

4 **Theorem 1.** *With the optimal critical network $D$ and the classifier $C$ fixed, the optimization of*
5 *generator $G$ is equivalent to minimize $\lambda \cdot d_W(\hat{\mathbb{P}}_N, \mathbb{Q}) - D_{KL}(\mathbb{Q}||\mathbb{P}_c)$.*

6 *Proof.* Recall the objective function defined in Eq. (1),

$$\min_G \max_{D,C} U(C, G, D) = \lambda\big(\mathbb{E}_{\hat{\mathbb{P}}_N}[D(\boldsymbol{x}, y)] - \mathbb{E}_{\mathbb{Q}}[D(\boldsymbol{x}, y)]\big) - \mathbb{E}_{\mathbb{Q}}[\ell(C(\boldsymbol{x}), y)]. \tag{1}$$

7 Given the optimal critical network $D$ and classifier $C$, the generator $G$ is optimized by minimizing
8 the function

$$V_{C,D}(G) = \lambda\big(\mathbb{E}_{\hat{\mathbb{P}}_N}[D(\boldsymbol{x}, y)] - \mathbb{E}_{\mathbb{Q}}[D(\boldsymbol{x}, y)]\big) - \mathbb{E}_{\mathbb{Q}}[\ell(C(\boldsymbol{x}), y)]. \tag{2}$$

9 As the critical network $D$ is optimized for describe the Wasserstein, which means that

$$\mathbb{E}_{\hat{\mathbb{P}}_N}[D(\boldsymbol{x}, y)] - \mathbb{E}_{\mathbb{Q}}[D(\boldsymbol{x}, y)] = d_W(\hat{\mathbb{P}}_N, \mathbb{Q}). \tag{3}$$

10 Then we consider the last term in Eq. (2),

$$\begin{aligned}
\mathbb{E}_{\mathbb{Q}}[\ell(C(\boldsymbol{x}), y)] &= \mathbb{E}_{\mathbb{Q}}[-\log p_c(y|\boldsymbol{x})] \\
&= \int p_g(\boldsymbol{x}, y) \log \frac{p_g(\boldsymbol{x}, y)}{p_c(\boldsymbol{x}, y) p_g(y|\boldsymbol{x})} \mathrm{d}(\boldsymbol{x}, y) \\
&= \int p_g(\boldsymbol{x}, y) \log \frac{p_g(\boldsymbol{x}, y)}{p_c(\boldsymbol{x}, y)} + p_g(\boldsymbol{x}, y) \log \frac{1}{p_g(y|\boldsymbol{x})} \mathrm{d}(\boldsymbol{x}, y) \\
&= D_{KL}(p_g(\boldsymbol{x}, y)||p_c(\boldsymbol{x}, y)) + H_g(y|\boldsymbol{x}).
\end{aligned} \tag{4}$$

11 Note that the label $y$ is provided to $G$ durning generation progress. As a result, $H_g(y|\boldsymbol{x})$ is irrelevent
12 to $G$. By concreting Eq. (4) and Eq. (3), The proof of Theorem 1 is completed. $\qquad\square$

### 13 A.2 Proof of Theorem 2

14 **Theorem 2.** *Consider $\boldsymbol{x}$ as the input samples of classifier $C$, and the distribution $\mathbb{Q} \in \mathbb{B}_\epsilon(\hat{\mathbb{P}}_N)$ lays*
15 *in a Wasserstein ball centered at $\hat{\mathbb{P}}_N$ with radius $\epsilon$. Then for any $\epsilon \geq 0$ and $\alpha \geq 1 + \beta$, we have*

$$\epsilon \|\nabla_{\boldsymbol{z}} \ell(\boldsymbol{z})\|_{\hat{\mathbb{P}}_N}^{\alpha*} - \epsilon^{\beta+1} \|h(\boldsymbol{z})\|_{\hat{\mathbb{P}}_N}^{\overline{\frac{\alpha}{\alpha-\beta-1}}} \leq \mathbb{E}_{\mathbb{Q}}(\ell(\boldsymbol{z})) - \mathbb{E}_{\hat{\mathbb{P}}_N}(\ell(\boldsymbol{z})) \leq \epsilon \|\nabla_{\boldsymbol{z}} \ell(\boldsymbol{z})\|_{\hat{\mathbb{P}}_N}^{\alpha*} + \epsilon^{\beta+1} \|h(\boldsymbol{z})\|_{\hat{\mathbb{P}}_N}^{\alpha*}, \tag{5}$$

16 *where $\|f(\boldsymbol{z})\|_{\hat{\mathbb{P}}_N}^{\alpha} \triangleq (\frac{1}{N} \sum_{i=1, \boldsymbol{z} \sim \hat{\mathbb{P}}_N}^{N} (\|f(\boldsymbol{z}_i)\|^\alpha))^{1/\alpha}$, $\alpha_* = \frac{\alpha}{\alpha-1}$, and $h(\boldsymbol{z})$ is a function and $\beta \in$*
17 *$(0, 1]$ a constant which satisfy $\|\nabla_{\boldsymbol{z}} \ell(\boldsymbol{z}_1) - \nabla_{\boldsymbol{z}} \ell(\boldsymbol{z}_2)\| \leq h(\boldsymbol{z}_2) \cdot \|\boldsymbol{z}_1 - \boldsymbol{z}_2\|^\beta$ for any $\boldsymbol{z} = (\boldsymbol{x}, y) \in \mathcal{Z}$.*

18 In this part, we firstly proof the right part of this theorem which is an upper bound of $\mathbb{E}_{\mathbb{Q}}(\ell(\boldsymbol{z})) -$
19 $\mathbb{E}_{\hat{\mathbb{P}}_N}(\ell(\boldsymbol{z}))$. Then we provide the proof of a lower bound of it. By combining them, the proof of
20 theorem 2 is completed.

21 *Proof.* Considering the inner part of the proposed objective function defined as follows,

$$\sup_{\mathbb{Q}\in\mathbb{B}_\epsilon(\hat{\mathbb{P}}_N)} \mathbb{E}_{\mathbb{Q}}[\ell_\theta(\boldsymbol{z})] = \inf_{\lambda\geq 0} \lambda\epsilon + \sup_{\mathbb{Q}\in\mathcal{M}(\mathcal{Z})} \int_{\mathcal{Z}} \ell_\theta(\boldsymbol{z})\mathbb{Q}\left(\mathrm{d}(\boldsymbol{z})\right) - \lambda \cdot d(\mathbb{Q}, \hat{\mathbb{P}}_N). \tag{6}$$

22 Under the situation that the network $D$ is optimized for calculating the Wasserstein distance, we
23 consider the network $D$ is sufficient to describe the Wasserstein distance. Recall that we define the
24 Wasserstein distance as

$$d_W(\mathbb{Q}_1, \mathbb{Q}_2) \triangleq \min_{\Pi\in\mathcal{M}(\Pi)} \left\{ \int_{\mathcal{Z}\times\mathcal{Z}} s(\boldsymbol{z}_1, \boldsymbol{z}_2)\Pi\left(\mathrm{d}\boldsymbol{z}_1, \mathrm{d}\boldsymbol{z}_2\right) \right\}. \tag{7}$$

25 Assuming the metric $s(\cdot, \cdot)$ is induced by some norm $\|\cdot\|^\alpha$, it is easy to be reformulated as follows:

$$d_W(\mathbb{Q}_1, \mathbb{Q}_2) = \min_{\Pi\in\mathcal{M}(\Pi)} \left\{ \int_{\mathcal{Z}\times\mathcal{Z}} \|\boldsymbol{z}_1 - \boldsymbol{z}_2\|^\alpha \Pi\left(\mathrm{d}\boldsymbol{z}_1, \mathrm{d}\boldsymbol{z}_2\right) \right\}. \tag{8}$$

26 Plugging Eq. (8) into Eq. (6) gives us

$$\mathbb{E}_{\mathbb{Q}}(\ell(\boldsymbol{z})) = \inf_{\lambda\geq 0} \left\{ \lambda\epsilon + \frac{1}{n}\sum_{i=1}^n \sup_{\boldsymbol{z}\in\mathcal{Z}} \left(\ell(\boldsymbol{z}) - \lambda\|\boldsymbol{z} - \boldsymbol{z}'\|^\alpha\right) \right\} \tag{9}$$

27 where $\boldsymbol{z} \sim \mathbb{Q}$ and $\boldsymbol{z}' \sim \hat{\mathbb{P}}_N$, then we consider a upper bound that

$$\sup_{\boldsymbol{z}\in\mathcal{Z}}\{\ell(\boldsymbol{z}) - \ell(\boldsymbol{z}_i') - \lambda \cdot \|\boldsymbol{z} - \boldsymbol{z}_i'\|^\alpha\}$$

$$\leq \sup_{\boldsymbol{z}\in\mathcal{Z}}\{\|\nabla_{\boldsymbol{z}}\ell(\boldsymbol{z}_i')\|_* \cdot \|\boldsymbol{z} - \boldsymbol{z}_i'\| + h(\boldsymbol{z}_i') \cdot \|\boldsymbol{z} - \boldsymbol{z}_i'\|^{\beta+1} - \lambda \cdot \|\boldsymbol{z} - \boldsymbol{z}_i'\|^\alpha\}$$

$$\leq \sup_{\boldsymbol{z}\in\mathcal{Z}}\{\|\nabla_{\boldsymbol{z}}\ell(\boldsymbol{z}_i')\|_* \cdot \|\boldsymbol{z} - \boldsymbol{z}_i'\| + h(\boldsymbol{z}_i') \cdot \|\boldsymbol{z} - \boldsymbol{z}_i'\|^{\beta+1} - \lambda \cdot \|\boldsymbol{z} - \boldsymbol{z}_i'\|^\alpha + C \cdot \|\boldsymbol{z} - \boldsymbol{z}_i'\|^{\gamma+1}\}$$

$$\leq \sup_{\xi\geq 0}\{\|\nabla_{\boldsymbol{z}}\ell(\boldsymbol{z}_i')\|_* \cdot \xi + h(\boldsymbol{z}_i') \cdot \xi^{\beta+1} + C \cdot \xi^{\gamma+1} - \lambda \cdot \xi^\alpha\}, \tag{10}$$

28 where $0 \leq C, 1 < \gamma < \beta$ and $\xi := \|\boldsymbol{z} - \boldsymbol{z}_i'\|$. Following Young's inequality for products that
29 $ab \leq \frac{a^p}{p} + \frac{b^q}{q}$, we set $p = \frac{\alpha-1}{\alpha-1-\beta}, q = \frac{\alpha-1}{\beta}$ satisifing $\frac{1}{p} + \frac{1}{1} = 1$ and $a = \varphi^{1/p}\xi^{1/p}, b = \varphi^{-1/q}\xi^{\alpha/q}$.
30 Then for any $t > 0$ and $\varphi > 0$, it holds that

$$\xi^{\gamma+1} \leq \xi^{\beta+1} \leq \xi^{\alpha+1} \leq \frac{\alpha-1-\beta}{\alpha-1}\varphi\xi + \frac{\beta}{\alpha-1}\varphi^{-\frac{\alpha-1-\beta}{\beta}}\xi^\alpha. \tag{11}$$

31 Replacing $\xi^{\gamma+1}$ and $\xi^{\beta+1}$ with the last term of Eq. (11), it gives us

$$\sup_{\xi\geq 0}\{\|\nabla_{\boldsymbol{z}}\ell(\boldsymbol{z}_i')\|_* \cdot \xi + h(\boldsymbol{z}_i') \cdot \xi^{\beta+1} + C \cdot \xi^{\gamma+1} - \lambda \cdot \xi^\alpha\}$$

$$\leq \sup_{\xi\geq 0}\{(\|\nabla_{\boldsymbol{z}}\ell(\boldsymbol{z}_i')\|_* + \frac{\alpha-\beta-1}{\alpha-1} \cdot h(\boldsymbol{z}_i') \cdot \varphi_1 + \frac{\alpha-\gamma-1}{\alpha-1} \cdot C \cdot \varphi_2) \cdot \xi$$

$$- (\lambda - \frac{\beta}{\alpha-1} \cdot h(\boldsymbol{z}_i') \cdot \varphi_1^{-\frac{\alpha-\beta-1}{\beta}} - \frac{\gamma}{\alpha-1} \cdot C \cdot \varphi_2^{\frac{\alpha-\gamma-1}{\gamma}}) \cdot \xi^\alpha\}$$

$$\leq \sup_{\xi\geq 0}\left\{\mathcal{G}_\varphi(\boldsymbol{z}_i') \cdot \xi - (\lambda - \mathcal{N}_\varphi) \cdot \xi^\alpha\right\}, \tag{12}$$

32 where $\mathcal{G}_\varphi(\boldsymbol{z}_i') = \|\nabla_{\boldsymbol{z}}\ell(\boldsymbol{z}_i')\|_* + \frac{\alpha-\beta-1}{\alpha-1} \cdot h(\boldsymbol{z}_i') \cdot \varphi_1 + \frac{\alpha-\gamma-1}{\alpha-1} \cdot C \cdot \varphi_2$ and $\mathcal{N}_\varphi = \lambda - \frac{\beta}{\alpha-1} \cdot h(\boldsymbol{z}_i') \cdot$
33 $\varphi_1^{-\frac{\alpha-\beta-1}{\beta}} - \frac{\gamma}{\alpha-1} \cdot C \cdot \varphi_2^{\frac{\alpha-\gamma-1}{\gamma}}$. Considering the value of Eq. (12) is $+\infty$ when $\lambda \leq \mathcal{N}_\varphi$, we solve
34 Eq. (12) over $\xi$ and conclude that

$$\mathbb{E}_{\mathbb{Q}}(\ell(\boldsymbol{z})) - \mathbb{E}_{\hat{\mathbb{P}}_N}(\ell(\boldsymbol{z}))$$

$$\leq \inf_{\lambda\geq\mathcal{N}_\varphi} \left\{\lambda\epsilon^\alpha + \alpha^{-\frac{\alpha}{1-\alpha}}(\alpha-1)(\lambda-\mathcal{N}_\varphi)^{-\frac{1}{\alpha-1}}(\|\mathcal{G}_\varphi\|_{\hat{\mathbb{P}}_N}^{\frac{\alpha}{\alpha-1}})^{\frac{\alpha}{\alpha-1}}\right\}$$

$$\leq \epsilon\|\mathcal{G}_\varphi\|_{\hat{\mathbb{P}}_N}^{\frac{\alpha}{\alpha-1}} + \mathcal{N}_\varphi\epsilon^\alpha. \tag{13}$$

Plugging $\mathcal{G}_\varphi$ and $\mathcal{N}_\varphi$ into Eq. (13) and solving the minimization problem on $\varphi$, we obtain the right part of Theorem 2. Next we step to the left part of Theorem 2. We firstly began with a lower bound of $\sup_{\mathbb{Q}\in\mathbb{B}_\epsilon(\hat{\mathbb{P}}_N)} \mathbb{E}_{\mathbb{Q}}(\ell(\boldsymbol{z})) - \mathbb{E}_{\hat{\mathbb{P}}_N}(\ell(\boldsymbol{z}))$ as follows:

$$
\sup_{\boldsymbol{z}_i \in \mathcal{Z}} \Big\{ \frac{1}{N} \sum_{i=1}^N [\ell(\boldsymbol{z}_i) - \ell(\boldsymbol{z}_i')] : \big(\frac{1}{N} \sum_{i=1}^N \|\boldsymbol{z}_i - \boldsymbol{z}_i'\|^\alpha\big)^{\frac{1}{\alpha}} \le \epsilon \Big\}
$$

$$
\ge \sup_{\boldsymbol{z}_i \in \mathcal{Z}} \Big\{ \frac{1}{N} \sum_{i=1}^N \big[ \nabla_{\boldsymbol{z}} \ell(\boldsymbol{z}_i') \|\boldsymbol{z}_i - \boldsymbol{z}_i'\| - h(\boldsymbol{z}_i')\|\boldsymbol{z}_i - \boldsymbol{z}_i'\|^{\beta+1} \big] : \big(\frac{1}{N} \sum_{i=1}^N \|\boldsymbol{z}_i - \boldsymbol{z}_i'\|^\alpha\big)^{\frac{1}{\alpha}} \le \epsilon \Big\}
$$

$$
\ge \sup_{\boldsymbol{z}_i \in \mathcal{Z}} \Big\{ \frac{1}{N} \sum_{i=1}^N \nabla_{\boldsymbol{z}} \ell(\boldsymbol{z}_i') \|\boldsymbol{z}_i - \boldsymbol{z}_i'\| : \big(\frac{1}{N} \sum_{i=1}^N \|\boldsymbol{z}_i - \boldsymbol{z}_i'\|^\alpha\big)^{\frac{1}{\alpha}} \le \epsilon \Big\}
$$
$$
- \sup_{\boldsymbol{z}_i \in \mathcal{Z}} \Big\{ \frac{1}{N} \sum_{i=1}^N h(\boldsymbol{z}_i')\|\boldsymbol{z}_i - \boldsymbol{z}_i'\|^{\beta+1} : \big(\frac{1}{N} \sum_{i=1}^N \|\boldsymbol{z}_i - \boldsymbol{z}_i'\|^\alpha\big)^{\frac{1}{\alpha}} \le \epsilon \Big\}
$$
(14)

Further we conclude that with the help of Holder's inequality,

$$
\sup_{\boldsymbol{z}_i \in \mathcal{Z}} \Big\{ \frac{1}{N} \sum_{i=1}^N \nabla_{\boldsymbol{z}} \ell(\boldsymbol{z}_i') \|\boldsymbol{z}_i - \boldsymbol{z}_i'\| : \big(\frac{1}{N} \sum_{i=1}^N \|\boldsymbol{z}_i - \boldsymbol{z}_i'\|^\alpha\big)^{\frac{1}{\alpha}} \le \epsilon \Big\}
$$
$$
= \sup_{\xi \in \mathbb{R}} \Big\{ \frac{1}{N} \sum_{i=1}^N \nabla_{\boldsymbol{z}} \ell(\boldsymbol{z}_i') \xi_i : \big(\frac{1}{N} \sum_{i=1}^N \xi_i^\alpha\big)^{\frac{1}{\alpha}} \le \epsilon \Big\}
$$
$$
= \epsilon \|\nabla_{\boldsymbol{z}} \ell(\boldsymbol{z})\|_{\hat{\mathbb{P}}_N}^{\alpha_*}.
$$
(15)

Wee also have that

$$
\sup_{\boldsymbol{z}_i \in \mathcal{Z}} \Big\{ \frac{1}{N} \sum_{i=1}^N h(\boldsymbol{z}_i')\|\boldsymbol{z}_i - \boldsymbol{z}_i'\|^{\beta+1} : \big(\frac{1}{N} \sum_{i=1}^N \|\boldsymbol{z}_i - \boldsymbol{z}_i'\|^\alpha\big)^{\frac{1}{\alpha}} \le \epsilon \Big\}
$$
$$
= \sup_{\xi \in \mathbb{R}} \Big\{ \frac{1}{N} \sum_{i=1}^N \nabla_{\boldsymbol{z}} h(\boldsymbol{z}_i') \xi_i^{\beta+1} : \big(\frac{1}{N} \sum_{i=1}^N \xi_i^\alpha\big)^{\frac{1}{\alpha}} \le \epsilon \Big\}
$$
$$
= \epsilon^{\beta+1} \|h(\boldsymbol{z})\|_{\hat{\mathbb{P}}_N}^{\frac{\alpha}{\alpha-\beta-1}}.
$$
(16)

The proof is completed. $\qquad\square$

## A.3   Proof of Theorem 3

**Theorem 3.** *For any $0 < \delta < 1$, with probability at least $1 - \delta$ with respect to the sampling,*

$$
\mathbb{E}(\ell(C(\boldsymbol{x}, \boldsymbol{\theta}), y)) \le \mathbb{E}_{\mathbb{Q}}(\ell(C(\boldsymbol{x}, \boldsymbol{\theta}), y)) + \frac{12\sqrt{R}}{n}\big(\log \frac{n}{3\sqrt{R}} + 1\big) + \sqrt{\frac{8\log(2/\delta)}{N}},
$$
(17)

*and for any $\zeta > \frac{12\sqrt{R}}{n}\big(\log \frac{n}{3\sqrt{R}} + 1\big) + \sqrt{\frac{8\log(2/\delta)}{N}}$, we have*

$$
P\left(\ell(C(\boldsymbol{x}, \boldsymbol{\theta}), y) \ge \mathbb{E}_{\mathbb{Q}}(\ell(C(\boldsymbol{x}, \boldsymbol{\theta}), y)) + \zeta\right) \le \frac{\mathbb{E}_{\mathbb{Q}}(\ell(C(\boldsymbol{x}, \boldsymbol{\theta}), y)) + \frac{12\sqrt{R}}{n}\big(\log \frac{n}{3\sqrt{R}} + 1\big) + \sqrt{\frac{8\log(2/\delta)}{N}}}{\mathbb{E}_{\mathbb{Q}}(\ell(C(\boldsymbol{x}, \boldsymbol{\theta}), y)) + \zeta}.
$$
(18)

*where $R$ is only related to the architecture of the neural network.*

*Proof.* As description in the Theorem 8 in [2], for any integer $N$ and $\delta \in (0, 1)$, the risk bounds can be written as that

$$
\mathbf{E}\mathcal{L}(Y, f(X)) \le \hat{\mathbf{E}}_N \phi(Y, f(X)) + \mathfrak{R}_N(\tilde{\phi} \circ F) + \sqrt{\frac{8\ln(2/\delta)}{N}}.
$$
(19)

47 Setting the loss function $\mathcal{L}$ and $Y$ as $\mathcal{L}(\boldsymbol{x}, y) = \phi(\boldsymbol{x}, y) = \ell(C(\boldsymbol{x}, \boldsymbol{\theta}), y)$, it yields that

$$\mathbb{E}(\ell(C(\boldsymbol{x}, \boldsymbol{\theta}), y)) \leq \mathbb{E}_{\mathbb{Q}}(\ell(C(\boldsymbol{x}, \boldsymbol{\theta}), y)) + 2\mathfrak{R}_N(\ell(C(\boldsymbol{x}, \boldsymbol{\theta}), y)) + \sqrt{\frac{8\ln(2/\delta)}{N}}, \quad (20)$$

48 We use Lemma A.8 in [1] and get the Rademacher complexity estimate of neural networks $C$ with
49 loss function $\ell$ as follows,

$$\mathfrak{R}_N(\ell(C(\boldsymbol{x}, \boldsymbol{\theta}), y)) = \frac{12\sqrt{R}}{n}(\log\frac{n}{3\sqrt{R}} + 1), \quad (21)$$

50 where $R$ is only related with the architecture of neural network and defined as follows and the detailed
51 notation can be find in [1],

$$R = \frac{4B^2 \ln\left(2W^2\right)}{\gamma^2 \epsilon^2} \left(\prod_{j=1}^{L} s_j^2 \rho_j^2\right) \left(\sum_{i=1}^{L} \left(\frac{b_i}{s_i}\right)^{2/3}\right)^3. \quad (22)$$

52 Plugging result in Eq. (21) into Eq. (20) drives the first part of Theorem 3. Further, we can easily
53 obtain the second part of Theorem 3 by applying Markov's inequality here to obtain,

$$P\left(\ell(C(\boldsymbol{x}, \boldsymbol{\theta}), y) \geq \mathbb{E}_{\mathbb{Q}}(\ell(C(\boldsymbol{x}, \boldsymbol{\theta}), y)) + \zeta\right) \leq \frac{\mathbb{E}(\ell(C(\boldsymbol{x}, \boldsymbol{\theta}), y))}{\mathbb{E}_{\mathbb{Q}}(\ell(C(\boldsymbol{x}, \boldsymbol{\theta}), y)) + \zeta}$$

$$\leq \frac{\mathbb{E}_{\mathbb{Q}}(\ell(C(\boldsymbol{x}, \boldsymbol{\theta}), y)) + \frac{12\sqrt{R}}{n}(\log\frac{n}{3\sqrt{R}} + 1) + \sqrt{\frac{8\log(2/\delta)}{N}}}{\mathbb{E}_{\mathbb{Q}}(\ell(C(\boldsymbol{x}, \boldsymbol{\theta}), y)) + \zeta} \quad (23)$$

54 which completes the proof. □

## A.4 Proof of Theorem 4

56 **Theorem 4.** *For the generator $G$ and classifier $C$ fixed, the optimal discriminator $D$ is*

$$D^*_{G,C}(\boldsymbol{x}, y) = \frac{p_{data}(\boldsymbol{x}, y)}{p_{data}(\boldsymbol{x}, y) + p_g(\boldsymbol{x}, y)}, \quad (24)$$

57 *where $p_g(\boldsymbol{x})$ is the distribution generated by $G$.*

58 *Proof.* Given the generator and classifier, the loss function can be written as

$$V(D) = \int p_{data}(\boldsymbol{z}) \log D(\boldsymbol{z}) \mathrm{d}\boldsymbol{z} + \int p_g(\boldsymbol{z}) \log(1 - D(\boldsymbol{z})) \mathrm{d}\boldsymbol{z}$$

$$= \int p_d(\boldsymbol{z}) \log D(\boldsymbol{z}) + p_g(\boldsymbol{z}) \log(1 - D(\boldsymbol{z})) \mathrm{d}\boldsymbol{z}. \quad (25)$$

59 Following the proof in GAN [3], the function $V(D)$ achieves its maximum at $\frac{p_{data}(\boldsymbol{z})}{p_{data}(\boldsymbol{z}) + p_g(\boldsymbol{z})}$. □

## A.5 Proof of Theorem 5

61 **Theorem 5.** *With the optimal discriminator $D$ and the classifier $C$ fixed, the optimization of generator*
62 *$G$ is equivalent to $-\log 4 + 2JSD(\hat{\mathbb{P}}_N||\mathbb{Q}) - 1/\lambda \cdot D_{KL}(\mathbb{Q}||\mathbb{P}_c)$.*

63 *Proof.* Following the conclusion obtained in Theorem 4, with the optimal $D^*_{G,C}(\boldsymbol{x}, y) =$
64 $\frac{p_{data}(\boldsymbol{x}, y)}{p_{data}(\boldsymbol{x}, y) + p_g(\boldsymbol{x}, y)}$, the minimac game for $G$ can be reformulated as:

$$V(G, C) = \int p_{data}(\boldsymbol{z}) \log \frac{p_{data}(\boldsymbol{z})}{p_{data}(\boldsymbol{z}) + p_g(\boldsymbol{z})} \mathrm{d}\boldsymbol{z} + \int p_g(\boldsymbol{z}) \log \frac{p_g(\boldsymbol{z})}{p_{data}(\boldsymbol{z}) + p_g(\boldsymbol{z})} \mathrm{d}\boldsymbol{z} + \lambda \int p_g(\boldsymbol{z}) \ell(\boldsymbol{z}) \mathrm{d}\boldsymbol{z}$$

$$= -\log 4 + 2JSD(p_{data}(\boldsymbol{z})||p_g(\boldsymbol{z})) \mathrm{d}\boldsymbol{z} + \lambda \int p_g(\boldsymbol{z})[-\log p_c(\boldsymbol{z})] \mathrm{d}\boldsymbol{z}$$

$$= -\log 4 + 2JSD(p_{data}(\boldsymbol{z})||p_g(\boldsymbol{z})) \mathrm{d}\boldsymbol{z} + \lambda(D_{KL}((p_g(\boldsymbol{z})||p_c(\boldsymbol{z})) + H_g(y|\boldsymbol{x})). \quad (26)$$

65 Noting that the label $y$ in $p_g(\boldsymbol{x}, y)$ is assigned during the generation, $H_g(y|\boldsymbol{x})$ is a constant
66 which is irrelevant with $G$. As a result, the generator $G$ will be optimized by $-\log 4 +$
67 $2JSD(p_{data}(\boldsymbol{z})||p_g(\boldsymbol{z})) \mathrm{d}\boldsymbol{z} + \lambda D_{KL}((p_g(\boldsymbol{z})||p_c(\boldsymbol{z}))$, which completes the proof. □

## B  Algorithm

---

**Algorithm 1** Proposed Method

---

**Input:** The batch size $m$, the loss balanced coefficient $\lambda$.

Initialize generator parameters $\theta_g$ for $G$, $\theta_d$ for $D$, and $\theta_c$ for $C$ with the training set $\{(\boldsymbol{x}_1, y_1), (\boldsymbol{x}_2, y_2), \ldots, (\boldsymbol{x}_N, y_N)\}$.

Sample a batch of pairs $(\boldsymbol{x}_g, y_g) \sim p_g(\boldsymbol{x}, y)$ and a batch of pairs $(\boldsymbol{x}_d, y_d) \sim p_d(\boldsymbol{x}, y)$.

Update $D$ by ascending along its gradients $\nabla_{\theta_d} \left[ \frac{1}{m} \left( \sum_{(\boldsymbol{x}_d, y_d)} D(\boldsymbol{x}_d, y_d) - \sum_{(\boldsymbol{x}_g, y_g)} D(\boldsymbol{x}_g, y_g) \right) \right]$

Update $G$ by ascending along its gradients $\nabla_{\theta_g} \left[ \frac{1}{m} \left( \lambda \cdot \sum_{(\boldsymbol{x}_g, y_g)} D(\boldsymbol{x}_g, y_g) - \sum_{(\boldsymbol{x}_d, y_d)} C(\boldsymbol{x}_g, y_g) \right) \right]$

Update $C$ by ascending along its gradients $\nabla_{\theta_c} \left[ \frac{1}{m} \left( \sum_{(\boldsymbol{x}_d, y_d)} \ell(C(\boldsymbol{x}_g, \boldsymbol{\theta}), y_g) \right) \right]$

**Output:** A distribution optimized classifier $C$ and a worst-case distribution generator $G$.

---