[Reviews · NeurIPS 2019]

Reviewer 1



This paper explores a very interesting idea, that unfortunately is marred by rushed writing. The paper is also filled with typos, that make understanding it and judging the quality of the work extremely hard. Exploring the central idea of the paper, that of a generator playing vs a discriminator and a classifier, and giving more intuition about this approach and why the authors expect it to work better than other traditional GAN based methods was extremely necessary. Instead the authors just mention it in passing just before section 3. This has the potential to be a good paper in a future version if all the writing issues are addressed and the paper is planned out a bit more. Nevertheless, I outline a few questions I had to the authors of the paper. 1. Why does the RHS of min max U(C,G,D) in the main equation (10) not depend on G? Same question for equation 14. 2. Is there a inf_{\lambda > 0} missing from the second term of the LHS of equation 4 and equation 5? If yes, how does that change things? 3. I understand trying to bound the error for the worst-case distribution in an \epsilon Wasserstein ball during the theoretical result, as that gives an upper bound for all other distributions. But why train on the worst-case distribution? What does the framework achieve from this? 4. Given that the generator approximates a worst-case distribution in the \epsilon ball, one would expect that for good data (without noise added) just the classifier would achieve better performance than this framework. Why isn't this the case in the MNIST set within the Imbalance 1 column and SVHN set under the Normal column?

Reviewer 2



The formulation in eq. 1 appears new. However, I am not convinced why minimizing the loss w.r.t. worst case distribution (in some epsilon ball) is a good idea (eq. 1). In this formulation, the task of classifier is made harder by having to model distributions that may be unlike the true distribution. Further, it is evident from experiments that when the epsilon is large, the method doesn’t work. Please comment on how the method is different from adversarial data augmentation methods for smaller epsilon. It is not clear to me how the second line of eq. 8 follows from first line and what Q_i are? Line 267, concludes that proposed method learns distributions closer to true distribution consistent with some claim in sec. 3. It is not clear which claim it is consistent with. Further, it is not clear why it would model “true” distribution. This is because, the generator is learned to model the worst case distribution within a epsilon ball of the empirical distribution and there is no reason for that worst case distribution to be the true distribution. In my view any closeness to true distribution is purely circumstantial. How does the method compare to data augmentation methods e.g. mixup or simple addition of noise to the training examples? This comparison is important because the proposed method can be viewed as a data augmentation method, where the generator acts as a learned data augmentor. Other minor points Please change “Crap” in the title and paper to something more prudent e.g. “Bad” Line 62: demotes -> denotes Line 67: assumed to include For clarity (e.g. in eq. 3) it may be better to use some other letter instead of “d” to represent d(Q, P) Line 89: matrix -> metric

Reviewer 3



I have studied machine learning for around eight years. This paper is out of my expertise. However, based my educated guess, this paper is well written. The theory seems strong and sound.

Reviewer 4



The paper has provided an interesting approach to conduct learning from crap training data. Different from existing three-player works that lie in semi-supervised setting, this paper aims to investigate the underlying data generating scheme with the help of generator, and thus a different three-player game has been established to learn from crap data. The authors have clearly illustrated the proposed algorithm. Beginning with a general optimization problem over probability distributions in the ambiguity set, the authors derived a tractable solution by introducing a generator to approximate the worst-case distribution and Wasserstein distance as the measure of distributions. With the help of a generator network, the proposed algorithm actually implies a robust solution that can deal with the worst-case data distribution. Experimental results on real world dataset demonstrate the effectiveness of exploring the generation and classification tasks in a unified framework. Most importantly, the paper provided some theoretical analysis to better understand the properties and optimization of the proposed algorithm.

[Author Response · NeurIPS 2019]

We thank reviewers for the constructive comments. We will include more intuition about the approach, improve the
presentation and fix all minor issues in the final version.

Re: Worst-case distribution. Minimizing the loss of the worst-case distribution $\mathbb{Q}$ implies an optimization over all
distributions within the ball of an appropriate radius $\epsilon$ (see Eq. (1)), which could also include the unknown real
distribution $\mathbb{P}_N$. Though the worst-case $\mathbb{Q}$ may not be exactly the real $\mathbb{P}_N$, *the classifier (i.e. $\theta$) must have fitted $\mathbb{P}_N$*
*better than (or equivalently with) $\mathbb{Q}$*, as the classification error over $\mathbb{Q}$ is the worst. In iterations, the worst-case $\mathbb{Q}$ will
be dynamically determined by the classifier, and the classifier will fit the real $\mathbb{P}_N$ increasingly better in an implicit way.

**To #R1.** Re: Reasons of working better. First, generators in existing methods tend to fit the empirical distribution.
Given a bad training set, their generated data could be worse. Second, these generators often produce "easy" samples
by cooperating with the classifier, and a nearly duplicate copy of the given bad training data could be sufficient but will
be useless to estimate the real data distribution. In contrast, our generator plays against the classifier and the capability
of the classifier can be largely enhanced over a distribution ball.

Re: G in Eqs. (10) and (14). RHS of Eqs. (10) and (14) indeed depends on G. The last two terms are expectations over
the distribution Q, which is approximated by G(z), as explained in line 93 of the paper.

Re: $\inf_{\lambda>0}$ in Eqs. (4)-(5). In Eq. (4), $\inf_{\lambda>0}$ is over both two terms of RHS. A pair of braces will be used to avoid
misleading. $\lambda$ in Eq. (5) indicates the corresponding optimal for $\inf_{\lambda>0}$, and it will change with the bound $\epsilon$ in Eq. (3).
Since $\epsilon$ is unknown, it is common to take $\lambda$ as a hyper parameter to be tuned in experiments (*e.g.* Theorem 1 in [1]).

Re: Good data. This comment is insightful. Though training data are clean, there still probably exists a *subtle* gap
between distributions of training and test data. Moreover, the generator could conduct "data augmentation" for the
classifier. We may thus receive a slightly better result, *e.g.* 99.42 v.s. 99.35 on MNIST under imbalance= 1.

**To #R3.** Re: Large $\epsilon$. A large $\epsilon$ leads to a set $\mathbb{B}_\epsilon$ of huge capacity (see Eq. (2)), which could be flooded with distributions
that are far away from both the empirical distribution $\hat{\mathbb{P}}_N$ and the real data distribution $\mathbb{P}_N$. It is therefore reasonable to
set $\epsilon$ within an appropriate range, as what we usually do with hyper-parameters in machine learning.

Re: Difference from adversarial data augmentation. Firstly, adversarial data augmentation usually aims to create adver-
sarial copies of training data by adding perturbations. In contrast, we generate new sample from a distribution. Secondly,
the amount of perturbation (pixels for image) is often constrained in classical methods, while we focus on a distribution
ball with a radius bound. In addition, Triple GAN and Triangle GAN tend to generates samples that can well fit the
classifier (*i.e.* cooperation between G and C), but we improve the classifier by challenging it with the generator.

Re: Eq. (8) and $\mathbb{Q}_i$. We more clearly express the second line of Eq. (8) as $\sup_{\mathbb{Q}_i} \frac{1}{N} \sum_{i=1}^{N} \int_{\mathcal{X}} \ell_\theta(\boldsymbol{x}, y) \mathbb{Q}_i(\mathrm{d}(\boldsymbol{x}, y)) - \lambda \cdot$
$\sup_{f \in \mathcal{F}} \left\{ \frac{1}{N} \sum_{i=1}^{N} \left[ f(\boldsymbol{x}_i, y_i) - \int_{\mathcal{X}} f(\boldsymbol{x}, y) \mathbb{Q}_i(\mathrm{d}(\boldsymbol{x}, y)) \right] \right\}$. $\mathbb{Q}_i$ is the conditional distribution of $(\boldsymbol{x}, y)$ given $(\boldsymbol{x}_i, y_i)$.
The joint distribution $\Pi$ of $(\boldsymbol{x}_i, y_i)$ and $(\boldsymbol{x}, y)$ with marginals $\hat{\mathbb{P}}_N$ and $\mathbb{Q}$ respectively (see Eq. (3)), can be written as
$\Pi = \frac{1}{N} \sum_{i=1}^{N} \delta_{(\boldsymbol{x}_i, y_i)} \otimes \mathbb{Q}_i$. According to the law of total probability, we can factorize $\mathbb{Q}$ as the first line of Eq. (8).

Re: True distribution. Sentences around Line 267 will be rephrased. Minimizing the worst-case expected loss implies
an optimization over all distributions in the $\epsilon$-ball where the real data distribution is also expected to be included.

Re: Comparison with data augmentation. The results shown in the first line of Table 1 are obtained by a combination
of randomly clip, horizon flip, and rotation. The second line is results of Mixup [2]. Table 1 shows that our method
outperforms both two comparison methods. Mixup also provides reasonable improvement but is not as outstanding as it
on regular datasets. Other GAN-based data augmentation methods have been included in Tables 1 and 2 of the paper.

**To #R4.** Thanks for your support. Source codes will be released.

**To #R5.** Re: Explanation of improvement. The algorithm receives per-
formance improvement, as the classifier can be implicitly optimized over
the real data distribution with the help of the worst-case distribution.
Our generator plays against with the classifier within a Wasserstein ball,
so that the capability of the classifier can be enhanced during the iterations.

Table 1: Accuracy (%) on CIFAR-10.

| Method | IF = 10 | G(0.2) | Reduced |
|---|---|---|---|
| Combination | 85.63 | 85.25 | 83.59 |
| Mixup [2] | 86.04 | 85.66 | 83.91 |
| Ours | **86.86** | **85.87** | **84.60** |

Re: Eq. (9). Thanks for your advice. We will improve Eq. (9) to make it clearer.

Re: JSD and WD. By replacing the critical network D in WGAN with a discriminator, we can easily obtain Eq. (14).
With the help of Theorems 4 and 5, the standard GAN framework can be view as a JSD version of framework defined in
Eq. (10). The difference between them is the distance metric used in Eq. (2).

[1] Kloft, M., Brefeld, U., Laskov, P., et al., 2009. Efficient and accurate lp-norm multiple kernel learning. NIPS.
[2] Zhang, H., Cisse, M., Dauphin, Y.N. and Lopez-Paz, D., 2017, Mixup: Beyond empirical risk minimization. ICLR.


[Meta-Review · NeurIPS 2019]

The authors propose a new training methodology where the model is learned to minimize the expected loss with respect to the worst case distribution within an epsilon ball of the empirical distribution. The adversarial formulation appears novel and the paper provides interesting theoretical results. The empirical evaluation of the proposed methods is well done and convincing. Hence, we recommend acceptance.